

# Screening for Snow/Snowmelt in SNPP VIIRS Aerosol Optical Depth Algorithm

Jingfeng Huang[1,2,3], Istvan Laszlo[3,4], Lorraine A. Remer[5], Hongqing Liu[3,6], Hai Zhang[3,6], Pubu Ciren[3,6], Shobha Kondragunta[3]

[1] Earth Resources Technology Inc., Laurel, MD, USA
[2] previously at Earth System Science Interdisciplinary Center (ESSIC)/Cooperative Institute for Climate and Satellites (CICS)-Maryland, University of Maryland, College Park, MD, USA
[3] National Oceanic and Atmospheric Administration, National Environmental Satellite, Data, and Information Service, Center for Satellite Applications and Research, College Park, MD, USA
[4] Department of Atmospheric and Oceanic Science, University of Maryland, College Park, Maryland, USA
[5] Joint Center for Earth Systems Technology, University of Maryland, Baltimore County, Baltimore, USA
[6] I. M. Systems Group, Inc., College Park, MD, USA

*Correspondence to:* Jingfeng Huang (Jingfeng.Huang@noaa.gov)

**Abstract.** The Visible/Infrared Imaging/Radiometer Suite (VIIRS) onboard the Suomi National Polar-orbiting Partnership (S-NPP) spacecraft provides validated daily global aerosol optical thickness (AOT) retrievals; however, a close examination of the VIIRS aerosol product identified residual snow and snowmelt contamination, resulting generally in an overestimation of AOT. The contamination was particularly evident over northern hemisphere high latitude regions during the spring thaw. To improve the product performance, we introduced a new empirical snow and snowmelt screening scheme that combines a Normalized Difference Snow Index (NDSI) and Brightness Temperature (BT) based snow test, a snow adjacency test and a spatial homogeneity test (aka. spatial filter). Testing of retrievals for May 18, 2014 indicated that compared to the previous, visible reflectance anomaly (VRA) based snow test, the new NDSI and BT based snow test screened out an additional 3.44% of VIIRS AOT retrievals, most of which were over high latitudes experiencing snowmelt. The new snow adjacency test and the homogeneity test degraded another 5.57% and 0.26%, respectively, otherwise 'Good' quality AOT retrievals. For the VIIRS vs. AERONET matchups over northern hemisphere high latitude regions during three years of spring (2013-2015), the new scheme also effectively screened out a significant number of the matchups that had anomalous high positive biases attributable to snow and snowmelt contamination. The new snow and snowmelt screening scheme was transferred to the Interface Data Processing Segment (IDPS) VIIRS aerosol algorithm on Jun 22, 2015. Subsequently no significant snow and snowmelt contamination was found during spring 2016. The scheme is also implemented in the new Enterprise VIIRS aerosol algorithm in the National Oceanic and Atmospheric Administration (NOAA) Enterprise Processing System (EPS) that became operational in 2017.

## 1 Introduction

Nowadays with increasing public awareness of air pollution and aerosol climatic effects, satellite observations of global aerosol loading and transport provide valuable information for improving our understanding of the impact of aerosols on weather, climate and public health (Kaufman et al., 2002; Quaas et al., 2008; Al-Saadi et al., 2005; Von Donkelaar et al., 2010; Kloog et al., 2011). Although satellite retrievals of aerosol optical properties in cloud-free and even in cloudy scenes have advanced tremendously over the past decades (Lenoble et al., 2013; Jethva et al., 2016; Meyer et al., 2015; Sayer et al., 2016), aerosol observations over snow scenes are still not feasible by passive sensors.

Most over-land aerosol algorithms from passive sensors make assumptions about surface reflectance properties in the retrieval scene in order to separate the signal from aerosol scattering in the atmosphere from the signal originating from reflectance from



Earth's surface (Levy et al., 2007; Jackson et al., 2013). Snow reflectance properties vary significantly with snow grain size (Wiscombe and Warren, 1980; Warren et al., 1982) and impurities in or on the snow (Doherty et al. 2010; Hadley and Kirchstetter, 2012), all of which change rapidly in time as the snow ages, and especially as it melts. The quickly changing optical properties of snow introduce too much uncertainty for global operational aerosol retrieval algorithms (Li et al., 2005). Therefore, screening out snow pixels is a necessary procedure in almost every aerosol retrieval algorithm, yet it remains a daunting challenge, particularly

if the scene is complicated with sub-pixel or melting snow. Therefore the aerosol optical thickness (AOT) retrievals adjacent to snow still have large uncertainties due to potential snow contamination (Lyapustin et al., 2012). Note that this snow contamination introduces a positive bias in the aerosol retrieval, as snow is bright in the visible and very dark in the near- and shortwave-infrared. Many passive satellite aerosol retrieval algorithms interpret the brighter-than-assumed visible surface reflectance as extra aerosol loading (Li et al., 2005). Even a small amount of sub-pixel snow in the retrieval scene can introduce artificial positive bias in the

AOT.

The Visible/Infrared Imaging/Radiometer Suite (VIIRS) aboard the Suomi National Polar-orbiting Partnership (S-NPP), launched on October 28, 2011, is the first satellite in the series of the United States' next generation polar-orbiting operational environmental satellite system, the Joint Polar Satellite System (JPSS). The VIIRS aerosol products, following the aerosol products from heritage sensors such as the Moderate Resolution Imaging Spectrometer (MODIS) and the Multiangle Imaging Spectroradiometer (MISR),

continue to succeed in providing daily global aerosol observations for operational and scientific user communities (Jackson et al., 2013; Liu et al., 2014; Huang et al., 2016; Zhang et al., 2016). By comparing the VIIRS aerosol products to the Aerosol Robotic Network (AERONET, Holben et al., 1998) ground measurements and the MODIS aerosol products, the validation results indicated that the AOT over land reached validated maturity on Jan 23, 2013 and the AOT over ocean reached validated maturity on May 2, 2012 (Liu et al., 2014; Huang et al., 2016).

With more than three years of the validated S-NPP VIIRS aerosol products publicly available and used in various user applications, further in-depth data analyses show that the VIIRS retrievals are consistently overestimating the AOT, when compared with AERONET, over high latitude regions in the northern hemisphere, especially during the spring thaw, when snow is melting. This implies potential snow and snowmelt contamination in the products.

The VIIRS aerosol products are generated operationally in the so-called Interface Data Processing Segment (IDPS), which is one

of the segments of the NPP project that processes the raw observations into environmental data records (geophysical parameters). Versions of the system, representing increasing maturity and algorithm updates, are referred to as Mx builds in a given block (In IDPS a block is similar to a Collection in the MODIS processing.) The original at-launch snow test, summarized in the next section, was implemented in the IDPS VIIRS aerosol algorithm from Dec 9, 2011 to Jun 22, 2015, until the Mx8.8 build in Block 1.x. The new snow and snowmelt mask described in this paper, was implemented in the Mx8.10 build on Jun 22, 2015 and has been running

in the IDPS ever since. The new snow mask was further refined by tuning threshold values, and it has been implemented in the National Oceanic and Atmospheric Administration (NOAA), Enterprise Processing System (EPS) VIIRS Aerosol Algorithm, which is replacing the IDPS algorithm.

This paper presents the identification of the contamination in Section 2, the development of a new snow and snowmelt screening scheme in Section 3, and the evaluation of the new scheme in the VIIRS operational aerosol products in Section 4, followed by

summary and discussion in Section 5.

Note that all snow screening tests discussed in this paper are designed to prevent the aerosol algorithm from making retrievals in inappropriate snow cover conditions. For a true snow product, users are directed to the S-NPP VIIRS Snow/Ice products (Key et al., 2013).





## 2 Background and Identification of the Problem

According to Jackson et al. (2013), the snow screening in the VIIRS land aerosol algorithm consists of two parts: the snow flag
from the upstream VIIRS Cloud Mask (VCM) product, and the internal snow test within the VIIRS aerosol algorithm itself. When
aerosol observations are feasible under daytime conditions, the VCM uses the VIIRS Gridded Snow Cover product in conjunction
with a reflectance based snow detection algorithm to check for snow surfaces. However, to avoid significant AOT overestimation
caused from snow pixels or sub-pixel snow conditions, the snow screening for aerosol observations needs to be stricter than the
general snow detection in the VCM. Thus an internal snow detection scheme within the aerosol algorithm becomes necessary to
complement the VCM snow flag. Prior to Jun 22, 2015, the internal snow detection in the operational IDPS VIIRS aerosol
algorithm is based on three tests: (1) the visible reflectance anomaly (VRA), (2) the ratio of top of the atmosphere (TOA) reflectance
in the 1240 nm channel ($\rho_{1240}$, VIIRS band M8) to the 865 nm channel ($\rho_{865}$, VIIRS band M7), and (3) the surface temperature
derived from the split window technique (Walton et al., 1998). This at-launch internal snow test is referred to as the 'VRA-based
snow test' hereafter in the paper. VRA is defined as:

$$\text{VRA} = \rho^s_{488} - 0.5 \times \rho^s_{672} \tag{1}$$

where $\rho^s_{488}$ and $\rho^s_{672}$ are 488 nm (VIIRS band M3) and 672 nm (VIIRS band M5) surface reflectances. Eq. 1 is based on the
retrieval assumption that the surface reflectance of the blue wavelength is roughly half of the red wavelength (Kaufman et al.,
1997; Jackson et al., 2013).  A significant deviation from this well-established surface reflectance relationship indicates a surface
outside of our range of assumptions and should not be used for aerosol retrievals.

In the VIIRS aerosol algorithm with the VRA-based snow test, as shown in Table 1, if the reflectances in the required bands are
available and the following conditions are met, the internal snow test sets the snow flag and no aerosol retrievals are reported in
the VIIRS aerosol product:

1.   VRA > 0.02,

2.   $\rho_{1240}/\rho_{865} < 0.9$,

3.   Surface temperature < 278 K,

4.   No cirrus, and

5.   Cloud mask is confidently or probably clear.

Applying this VRA-based snow test, the AOT retrievals over high latitude geographic regions were, however, consistently
overestimated during the boreal spring thaw, when snow was melting (Jackson et al., 2013; Liu et al., 2014; Huang et al., 2016).
Neither the external VCM snow tests nor the internal VRA-based snow test were able to effectively detect and screen out those
snow- and snowmelt-contaminated pixels, and anomalously high AOTs were reported as high quality retrievals. Those unfiltered
snow or snowmelt pixels are usually more reflective in the visible, and thus can result in falsely high AOT retrievals if they are
mistaken as aerosols. The global validation of the S-NPP VIIRS AOT product reported that such snow and snowmelt contamination
frequently happened during spring thaw over high latitude geographic regions, such as northern Canada and northern Russia. This
widespread residual snow and snowmelt contamination caused a significant high bias in the AOT product. Using the AERONET
ground measurement, the validation of the VIIRS AOT during Feb – May of 2013-2015 showed a strong positive bias of +0.073,
not meeting the requirement of ±0.06 when AOT≤0.8 (NOAA JPSS Level 1 Requirements Document,
http://www.jpss.noaa.gov/pdf/L1RDS_JPSS_REQ_1002_NJO_v2.10_100914_final-1.pdf). This required an alternative internal
snow test in the VIIRS aerosol algorithm to replace the VRA-based snow test.





An empirical snow detection technique using near infrared reflectances at 860 nm and 1240 nm and brightness temperature at 11 µm channel, was proposed by Li et al. (2005). This scheme was implemented in the MODIS operational aerosol algorithm beginning with Collection 5 (Levy et al., 2007, 2009, 2013). This paper explores the applicability of a similar approach in the VIIRS aerosol algorithm by using the VIIRS bands. Snow adjacency and spatial homogeneity tests complement this snow test to
form a systematic snow and snowmelt screening scheme in the VIIRS aerosol algorithm.

### 3. The New Snow and Snowmelt Screening Scheme

The new empirical snow and snowmelt screening scheme consists of three separate tests: snow test, snow adjacency test, and spatial homogeneity test.

### 3.1 The NDSI and BT based Snow Test

The snow test follows the approach of Li et al. (2005). Similar to step 3 in the VRA-based snow test this snow test also uses near-infrared and shortwave infrared TOA reflectances, but instead of their simple ratio it calculates the ratio of their difference to their sum, the so called Normalized Difference Snow Index (NDSI) (defined below) and compares it to a threshold value. Normalization is used as a means to adjust for the effects of the solar and view zenith angles because the normalized form is less sensitive to changes in the two angles than the simple ratio form (i.e. Walter-Shea et al., 1997). Again, similar to step 4 in the VRA-based
snow detection, it also uses a thermal-infrared brightness temperature test but with a different threshold value. In our adaptation we substitute the following VIIRS spectral bands for the MODIS bands in Li et al. (2005). We calculate the NDSI from $\rho_{865}$ and $\rho_{1240}$ (VIIRS bands M7 and M8), and the brightness temperature (BT) at 10.76 µm ($BT_{11\mu m}$, VIIRS band M15). To differentiate this test from the VRA-based snow test, we call this new snow test the "NDSI-based snow test" hereafter.

The theoretical basis of the NDSI-based snow test lies in the fact that the spectral dependence of snow reflectance is very different
from that of vegetation and soils. The reflectance of snow decreases rapidly with increasing wavelength from visible to shortwave infrared in the 800 – 1300 nm wavelength range due to strong ice absorption features centered near 1050 and 1240 nm. The reflectance of green vegetation decreases only slightly with increasing wavelength due to weak liquid water absorption bands centered near 960 and 1180 nm. The reflectances of soil and dry vegetation however typically increase with wavelength in the same spectral region. Therefore the reflectance of snow is higher than that of green vegetation and soil at visible bands but much
lower at shortwave infrared bands (Figure 3 therein Li et al. 2005). Based on the spectral properties of snow, vegetation, and soil, the NDSI, which was first used in Gao (1996), is defined as follows:

$$\text{NDSI} = \frac{(\rho_{865}-\rho_{1240})}{(\rho_{865}+\rho_{1240})} \tag{2}$$

where $\rho_{865}$ and $\rho_{1240}$ are reflectances at 865 nm (VIIRS M7) and 1240 nm (VIIRS M8) respectively. It is noteworthy that while the IDPS algorithm uses TOA reflectances in NDSI, the EPS algorithm uses Rayleigh-scattering and gas-absorption corrected
reflectances to minimize these effects on NDSI.

Similarly to Li et al. (2005), to avoid over-screening of vegetated pixels, $BT_{11}$ is also used for further stratification.

As Table 1 and the flowchart in Figure 1(a) show, this NDSI-based snow test will set a snow flag when the required reflectance and brightness temperature at respective bands are available and the following criteria are met:

1    NDSI > C1 (see below for threshold values used),
150 2    $BT_{11\mu m}$ < 285 K;
3    No cirrus, and



4    Cloud mask is confident or probably clear.

C1 has been set 0.01 in Mx8.10 and in newer versions of the IDPS Aerosol Algorithm. However, evaluation of retrievals over

pixels with heavy smog, particularly over Eastern China during the boreal spring season, showed that this threshold value resulted

in over-screening of such pixels. This happened because some heavy smog pixels also exhibited an NDSI data range (>0.01) and

lower brightness temperature ($BT_{11\mu m}<285K$) similar to those for snow pixels, indicating stronger absorption of smog particles in

the shortwave infrared than in the near infrared and relatively low brightness temperatures at $11\mu m$. Thus the criteria of NDSI >

0.01 prevented potentially good AOT retrievals of these China smog events (Huang et al., 2017). Adjusting the NDSI threshold

value showed that a higher threshold 0.10 helps to regain most AOT retrievals over the heavy air pollution pixels that previously

went missing in the IDPS algorithm. Therefore in the EPS algorithm the higher C1 value (0.1) has been adopted. At the same time,

the threshold in the spatial filter had to be adjusted as well (in Section 3.3) to ensure the new threshold values will not re-introduce

artifacts due to snow contamination in other areas (Huang et al., 2017).

Because snow is one of the conditions that prevent meaningful aerosol retrievals, there are no aerosol retrievals over pixels with

snow flags (Jackson et al., 2013). Therefore, in the VIIRS aerosol product, those pixels are filled with Fill Value and the quality

of the retrievals over those pixels is set as 'Not Produced'.

### 3.2 Snow Adjacency Test

Although the NDSI-based snow test improves the snow pixel detection significantly, some residual snowmelt contamination

surrounding the snow pixels still exists when we verify AOT retrievals over snow scenes. We attribute this contamination to

unidentified sub-pixel snow. To minimize such contamination, a snow adjacency test is introduced as an additional quality

assurance procedure in the aerosol algorithm. If the test shows a pixel is within an area of 7×7 pixels surrounding a snow pixel,

the aerosol retrieval for that pixel is likely susceptible to snowmelt contamination, and the condition becomes unfavorable for

meaningful aerosol retrievals.

As shown in Table 1 and Figure 1(b), the new snow adjacency test loops through the adjacent 7×7 pixels surrounding the central

snow pixel. For each of the 7×7 pixels except the central pixel, the snow adjacency quality flag is set if the following criteria are

met:

1    Center pixel is set as 'snow' over land,

2    No cirrus detected, and

3    Cloud Mask is confident or probably clear.

However, unlike the snow test, the snow adjacency test is designated as one of the 'Degradation' retrieval condition rather than a

180    'Not Produced' retrieval condition. After the snow adjacency quality flag is set for a particular pixel, the snow adjacency test

continues to check whether the aerosol retrieval quality is 'Good'. If the aerosol retrieval quality is 'Good', the snow adjacency

test degrades the aerosol retrieval quality from 'Good' to 'Degraded'; otherwise, the aerosol retrieval quality is not changed.

### 3.3 Spatial Homogeneity Test (Spatial Filter)

The third test, like the snow adjacency test, is meant to caution users that retrieved AOT may be susceptible to sub-pixel snow and

185    snowmelt situations. This alert becomes especially important during spring thaw when snowmelt pixels can spread over large

geographic areas. Pixels containing a surface with a mixture of exposed soil, vegetation, old snow and soggy slush introduce a

level of spatial inhomogeneity not seen during other seasons.  To identify this situation, an internal spatial homogeneity test is

introduced in the aerosol algorithm. The spatial filter calculates the standard deviation of the TOA 412 nm reflectance (□412,

VIIRS band M1) in a grouping of 3×3 pixels to assess the spatial homogeneity. If the assessment shows large spatial heterogeneity





190    of the M1 TOA reflectance within the surrounding 3×3 pixel area, the condition becomes unfavorable for meaningful aerosol retrievals. The 412-nm reflectance is used because of the generally low reflectance of snow-free land surfaces at this wavelength. As shown in Table 1 and Figure 1(c), the spatial filter sets the homogeneity test flag for the center pixel when the following criteria are met:

1    Aerosol retrieval flag of the center pixel is 'Good', and

2    The standard deviation of $\rho_{412}$ of 3×3 surrounding pixels exceeds C2.

C2 has been set to 0.05 in the Mx8.10 and newer versions of the IDPS Aerosol Algorithm. As it has been mentioned above, in the EPS algorithm the NDSI threshold was relaxed from its IDPS value to regain retrievals over heavy smog pixels (see Section 3.1), and consequently the value of C2 had to be adjusted accordingly to ensure a stricter homogeneity requirement to compensate. This is because smog events are usually homogeneous at these spatial scales, and thus an adjustment of the spatial filter threshold does

not cause over-screening of smog pixels or other retrieval scenes, yet manages to eliminate the spatially variable sub-pixel snow that appeared once the NDSI threshold was relaxed. The adjustment helps minimize the potential false aerosol retrievals over pixels with sub-pixel snow and NDSI values in the range of 0.01~0.10. However, to verify the new NDSI threshold will not introduce snow and snowmelt contamination, we closely examined 24 granules with smog pixels on smog prevalent days over China from Nov 29, 2015 to Feb 29, 2016, and another 30 granules with snow and snowmelt pixels over Canada on March 27-31 and May 28-

29, 2015. With same sets of threshold values of NDSI and spatial filter applied to both heavy smog and snow scenes, a new C2=0.004 was carefully selected to achieve a balance between screening for sub-pixel snow and allowing retrievals of heavy smog. More justification of the threshold changes can be found in Section 4 where the aerosol retrievals from the EPS algorithm are discussed.

Due to the nature of the spatial filter the homogeneity test not only helps detecting sub-pixel snow pixels but also helps screening

partial cloudy pixels that contain sub-pixel cloud. The test is particularly effective for those low level 'popcorn' cumulus clouds, which are so small in spatial scale causing higher spatial variability. Since both cloud and snow conditions are unfavorable conditions for meaningful satellite aerosol retrievals, the homogeneity test provides additional quality assurances to the VIIRS aerosol retrievals in terms of both snow and cloud screenings.

Similar to the snow adjacency test, the homogeneity test is designated as one of the 'Degradation' conditions rather than a 'Not

Produced' condition. However, for efficient aerosol retrieval the homogeneity test is only conducted when the aerosol retrieval quality of the center pixel is 'Good'. If this flag is 'Good' and the above criteria are met, the homogeneity test quality flag is set and the quality of the aerosol retrieval is degraded from 'Good' to 'Degraded'.

## 4 Evaluation of the new snow and snowmelt screening scheme

To demonstrate the effectiveness of the new snow and snowmelt screening scheme, we apply old and new schemes to the same

granule. In this test the new scheme uses the thresholds in the Mx8.10 IDPS algorithm (C1=0.01 and C2=0.05). As seen in Figure 2(a), an example of "good" quality VIIRS AOT retrievals clearly shows anomalously high AOT values over areas to the east and south of the Hudson Bay in Canada on May 19, 2015 when the snowpack was melting, as indicated by the snow cover map in Figure 2(c) from the NOAA National Centers for Environmental Information (NCEI) (https://www.ncdc.noaa.gov/snow-and-ice/snow-cover/us/20150519). Comparing the results of the new NDSI-based scheme in Figure 2(b) with the VRA-based snow test

of Figure 2(a) clearly shows that the amount of anomalously high AOT retrievals are significantly reduced, implying the new scheme has effectively reduced the snow and snowmelt contamination in the high quality AOT retrievals.



We can better understand the workings of the new scheme by comparing the spectral reflectance of different populations of pixels found in the granules of Figure 2. Five populations of pixels are shown in Figure 3: 1) snow pixels identified by the VRA-based snow test; 2) snow pixels identified by the NDSI-based snow test; 3) snow or snowmelt pixels flagged by the spatial filter; 4) snow

adjacent pixels flagged by the snow adjacency test; and 5) pixels with high quality aerosol retrievals from both versions of the algorithm, where the retrieval conditions should be less susceptible to snow and snowmelt contamination. The averages of TOA reflectances at 11 VIIRS spectral bands, the NDSI, and the BT11μm are calculated, respectively, for each population. The TOA reflectance $\rho_{865}$ and $\rho_{1240}$ are connected by a line segment to show the slopes better. Because ice particles have much stronger absorption at 1240 nm than soil and vegetation, the steeper the negative slope from 865 nm to 1240 nm, the higher the NDSI value,

the more likely snow exists in those pixels. While this slope is negative for snow and slightly positive for soil, as explained in Section 3.1, the pixels with sub-pixel snow should feature a reduced steepness of the slope depending on the relative coverage of soil and snow within the pixel. Similarly, because the absolute temperature of snow is generally lower than that of vegetation or soil, the more snow exists in the pixels, the lower the radiation emitted by the pixels, thus the lower the brightness temperature at 11μm (i.e. Gutman et al., 1995; Hori et al. 2006; Li et al., 2013). As shown in Figure 3, the NDSI and BT values for pixels identified

as snow by the VRA-based snow test are 0.260 and 265.8K, and those for pixels identified as snow by the NDSI-based snow test are 0.217 and 271.4K. Thus, the NDSI-based snow test has been more aggressive, categorizing pixels as snow despite their spectral signature and temperature being less characteristically snow-like (that is, darker and warmer) than was required by the VRA-based snow test. The snow adjacency and spatial filter tests continue in this vein identifying more pixels with even less snow-like characteristics: NDSI and $BT_{11μm}$ values are 0.017 and 283.2K for the spatial filter category and -0.041 and 283.5K for the snow

adjacency category. For the category with good quality aerosol retrieval after both schemes were applied, the NDSI is lowest (-0.071) and $BT_{11μm}$ is highest (288.4 K), indicating surface conditions with the least snow-like characteristics.

The VRA-based and NDSI-based snow screen schemes were implemented into the aerosol algorithm and applied globally for the testing date of May 18, 2014. Figure 4 shows the high quality VIIRS AOT retrievals with the old (Figure 4a) and new tests (Figure 4b), and their differences (Figure 4c). Most of the reduced high quality AOT retrievals were found over high latitude snowmelt

prevalent regions. They either became 'Not Produced' because of the stricter new snow test or were degraded from 'Good' to 'Degraded' by the new snow adjacency test and the spatial filter. To demonstrate the impact of the tests on the number of retrievals statistics of sample size and percentage change in the number of AOT retrievals were calculated for 18 10-degree latitude bins, and the results are shown in Figure 5. As expected, the new snow, adjacency and spatial homogeneity tests have largest impact over high latitude regions in the northern hemisphere, where snow is rapidly melting during boreal spring thaw. Overall, global

statistics indicate that the new snow test screened out an additional 3.44% VIIRS AOT retrievals, and the new snow adjacency test and the homogeneity test degraded another additional 5.57% and 0.26% 'Good' quality AOT retrievals, respectively. Although these percentages may change from day to day, they provide rough estimates of the magnitudes of the percentage change in the number of good aerosol retrievals when the VRA-based snow test is replaced by the new snow and snowmelt screening scheme.

The effectiveness of the new snow and snowmelt screening scheme is also verified from the validation of the VIIRS AOT products

with AERONET ground measurements. Figure 6 compares the VIIRS vs. AERONET matchups over [130ºW-50ºW, 50ºN-90ºN] during boreal spring Feb-May, 2013-2015, during which period the VRA-based snow test was still in operation in the VIIRS aerosol algorithm. A matchup is defined as the mean high quality VIIRS AOT retrievals within 27.5 km from an AERONET site and the mean AERONET AOT observations within ±30 minutes of the VIIRS overpass time. A significant number of anomalously high VIIRS AOT retrievals with positive biases were found in the retrievals plotted in Figure 6(a). Then for each matchup, the

corresponding NDSI, BT, snow adjacency and spatial standard deviations were calculated, so that we could apply the various tests of the NDSI-based snow screen using the different threshold values for the various versions of the algorithm. The VIIRS vs.



AERONET matchups in Figure 6(a) were removed if the new snow test would have prevented the retrieval or the new snow adjacency test or homogeneity test would have degraded the quality of the retrieval. Figure 6(b) and Figure 6(c) showed two screening conditions, one uses thresholds from the Mx8.10 IDPS algorithm and the other uses thresholds from the EPS algorithm.

Out of 260 matchups in Figure 6(a), using the IDPS thresholds, 43 were identified as snow, 94 were found with the adjacency test, and none with the spatial filter test. Because some of the snow pixels were also adjacent to other snow pixels, resulting in redundancy, 97 pixels were screened out, leaving 163 remaining matchups after the screening, with much improved accuracy, much lower uncertainty and much better agreement to AERONET. Similarly, the same level of improvement was also achieved with the EPS thresholds. 30 matchups were found with snow, 94 with the adjacency test and 81 with the spatial filter test because

of the tighter threshold for the standard deviation. Allowing for pixels with multiple conditions, the screening with the EPS thresholds resulted in 158 remaining matchups in Figure 6(c), having better agreement to AERONET. There are two anomalous points in Figure 6 (a)-(c) with higher positive biases in the red circles that were not screened out. With the same snow screening scheme, we ran the EPS algorithm on the same granule and found that the same two matchups with higher positive biases in the red circles were retrieved by the EPS algorithm as high quality retrievals but their biases were significantly reduced by ~0.2 from

0.5-0.6 to 0.3-0.4. This means the positive biases of these two anomalies seemed to be related to the AOT retrieval itself rather than the under-screening snow.

The new snow and snowmelt scheme was transferred to operation (TTO) in the Mx8.10 IDPS VIIRS aerosol algorithm on Jun 22, 2015 19:43 UTC. No significant snow and snowmelt contamination were found in the operational IDPS aerosol products during the 2016 spring thaw season, demonstrating the new scheme had improved the quality of the products with much better snow and

snowmelt screening. The new snow and snowmelt screening scheme is also implemented in the EPS VIIRS aerosol algorithm, but with its own thresholds (Laszlo and Liu, 2016). As previously discussed, the threshold values of the NDSI and spatial filter were adjusted in the EPS VIIRS aerosol algorithm to regain heavy smog retrievals while keeping the same level of snowmelt screening as in the Mx8.10 IDPS algorithm (Huang et al., 2017). Global assessment of the new tests in the EPS algorithm were conducted for the boreal spring season of 2015, which had significant snow and snowmelt contamination over northern hemisphere high

latitudinal regions (Figure 7a). The VIIRS aerosol retrieval in the spring thaw season of 2015, produced from the EPS algorithm in Figure 7(b), was compared to the operational IDPS products during the same season in Figure 7(a). Note that May 2015 preceded the implementation of the new snow scheme, which began in June of that year. Thus Figure 7(a) shows high AOT along the edge of the northern snow boundary, but Figure 7(b) with the new scheme does not. The new boreal spring seasonal VIIRS AOT retrievals are much improved in terms of the significantly reduced amount of anomalously high AOT values, particularly over

Northern Canada and Northern Russia. At the same time, Figure 7(b) also highlights that the EPS aerosol products regained more AOT retrievals of smog events over Eastern China during spring 2015, resulting in higher seasonal mean AOT there. This improvement results from the NDSI threshold adjustment. Other advantages to the EPS products, unrelated to the new snow mask, such as retrievals over the bright deserts can also be seen in Figure 7(b).

## 5 Summary and Discussion

Validation of the S-NPP VIIRS operational aerosol products revealed residual snow and snowmelt contaminations during the boreal spring thaw season over high latitude geographic regions. To reduce such contamination, we proposed a snow and snowmelt screening scheme that combines a new NDSI and BT based snow test, a snow adjacency test and a spatial filter based on the standard deviation of reflectance at 412 nm. The pixels flagged as snow by the snow test will become 'Not Produced', while the 'Good' AOT retrievals adjacent to identified snow pixels or with higher spatial heterogeneity will be 'Degraded'. It is noteworthy

that in the operational environment the cloud test should be conducted before the snow test because the snow detection requires





clear sky condition. The snow test should be conducted before the snow adjacency test and the spatial filter because the latter tests need to know whether the central pixel is identified as snow or whether the retrieval is a good quality one. The order of snow adjacency test and spatial filter however does not produce a difference in the aerosol retrieval. Since spatial filter requires standard deviation calculation, which is more computationally expensive than the snow adjacency test, it is usually arranged as the last test

of the snow and snowmelt scheme. The testing of the new scheme demonstrated significant improvements in the VIIRS aerosol retrievals with much fewer anomalous high AOT retrievals due to snow and snowmelt contamination.

A global testing on May 18, 2014, a typical day in spring thaw season when snow and snowmelt prevail, showed that the new snow test screened out an additional 3.44% 'Good' quality VIIRS AOT retrievals, which were otherwise contaminated by snow and snowmelt, and the new snow adjacency test and the homogeneity test degraded 5.57% and 0.26% 'Good' quality AOT

retrievals to 'Degraded' quality, respectively. Such percentage is expected to be lower in other seasons when AOT retrievals are expected to be less susceptible to snow and snowmelt contamination. In future works, in order to reach a more quantitative statistics for a better understanding of the relative contributions from each test, more testing dates at different seasons are needed.

The new snow and snowmelt scheme was also able to screen out significant number of the VIIRS vs. AERONET matchups that had anomalous high positive biases in the IDPS VIIRS aerosol products over Canada during the spring thaw season from 2013 to

2015. The new snow and snowmelt screening scheme was transferred to operation (TTO) in the operational IDPS VIIRS aerosol algorithm on Jun 22, 2015 19:43 UTC. No significant snow and snowmelt contamination was found in the operational IDPS aerosol products during the 2016 spring thaw season. The new scheme has also been implemented in the upcoming EPS aerosol algorithm, however with fine-tuned threshold values of NDSI and spatial filter tests, in order to regain some AOT retrievals over heavy smog events and to maintain the snow and snowmelt screening at the same strict level as in the IDPS algorithm. The VIIRS AOT

retrievals during spring 2015 produced by the EPS algorithm were much improved from the IDPS AOT product, in terms of the significantly reduced amount of anomalously high AOT values, particularly over Northern Canada and Northern Russia. The new EPS VIIRS aerosol algorithm became operational in July, 2017.

**Acknowledgements**

This study is supported by NOAA JPSS Program Office (Dr. Mitchell D. Goldberg, JPSS Program Scientist and Ms. Lihang Zhou, JPSS STAR Program Manager) under a grant to the Cooperative Institute for Climate and Satellites – Maryland (CICS-Maryland, Award # = NA14NES4320003, Title of Award = CICS: Cooperative Agreement 2014 – 2019). The manuscript contents are solely the opinions of the authors and do not constitute a statement of policy, decision, or position on behalf of NOAA or the U.S. government. All the operational IDPS VIIRS data used in this study are publicly accessible at the NOAA Comprehensive Large

Array-Data Stewardship System (CLASS, http://www.class.ngdc.noaa.gov/). The EPS VIIRS aerosol products used in this study are available from the VIIRS aerosol calibration and validation team upon request (http://www.star.nesdis.noaa.gov/smcd/emb/viirs_aerosol/index.php). For any questions related to the VIIRS aerosol dataset, please contact Istvan.Laszlo@noaa.gov or Shobha.Kondragunta@noaa.gov.





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



**Tables**

**Table 1. Criteria for the VRA based snow test, NDSI based snow test, snow adjacency test and homogeneity test**

| Tests | Old VRA Based Test | New NDSI Based Snow Test | Snow Adjacency Test | Spatial Filter |
|---|---|---|---|---|
| Criteria | 1. VRA > 0.02; <br> 2. $\rho_{1240}/\rho_{865} < 0.9$; <br> 3. Surface Temperature < 278 K; <br> 4. No cirrus, and <br> 5. Confidently or probably clear. | 1. NDSI > $C1$; <br> 2. $BT_{11\mu m} < 285$ K, <br> 3. No cirrus, and <br> 4. Confidently or probably clear. | For each of the 7×7 pixels except the central pixel, the snow adjacency quality flag is set if: <br> 1. Center pixel is set as 'snow' over land, <br> 2. No cirrus, and <br> 3. Confidently or probably clear. | Sets the homogeneity test flag for the center pixel If: <br> 1. Aerosol retrieval flag of the center pixel is 'Good', and <br> 2. The standard deviation of $\rho_{412}$ 3x3 surrounding pixels exceeds $C2$ |
| AOT Quality | Not Produced | Not Produced | Degraded to Medium | Degraded to Medium |
| Notes | 1. $VRA = \rho^s_{488} - 0.5 \times \rho^s_{672}$, $\rho^s_{488}$ and $\rho^s_{672}$ are 488 nm (VIIRS band M3) and 672 nm (VIIRS band M5) surface reflectance; <br> 2. ST derived from $BT_{11\,\mu m}$ and $BT_{12\mu m}$ | 1. $NDSI = \frac{(\rho_{865} - \rho_{1240})}{(\rho_{865} + \rho_{1240})}$, $\rho_{865}$ and $\rho_{1240}$ are reflectances at 865 nm (VIIRS M7) and 1240 nm (VIIRS M8) respectively; <br> 2. C1=0.01 for IDPS; C1=0.10 for EPS. | Check high quality AOT retrievals only | 1. Check high quality AOT retrievals at central pixel only; <br> 2. $\rho_{412}$ is reflectance at 412 nm (VIIRS M1); <br> 3. C2=0.05 for IDPS; C2=0.004 for EPS |



**Figures**

**(a)**

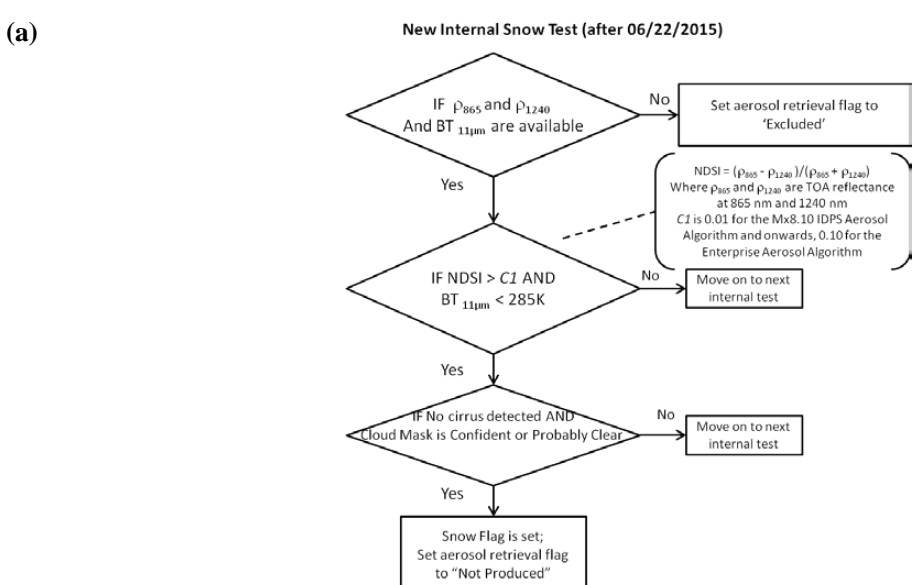

**(b)** **(c)**

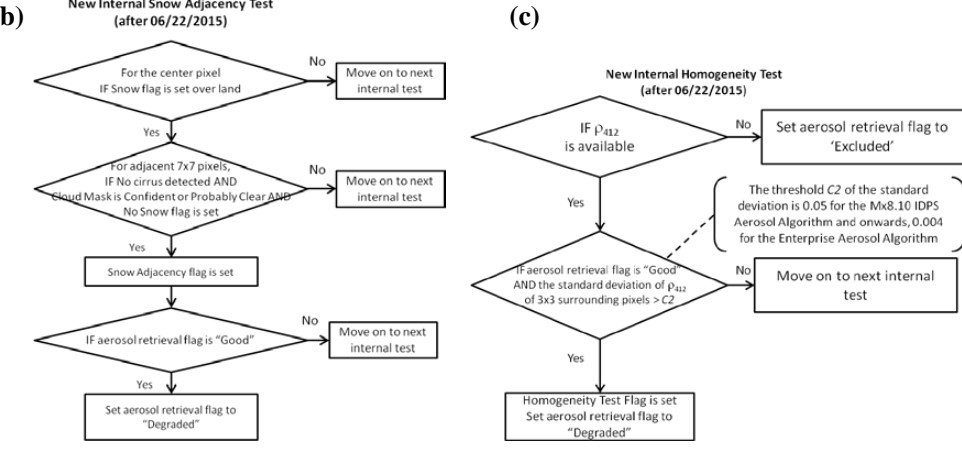


**Figure 1. Flow charts of the new snow and snowmelt snow screening scheme in the VIIRS aerosol algorithm: (a) internal snow test; (b) internal snow adjacency test; and (c) internal homogeneity test or spatial filter. The tests were transferred to operation to the Mx8.10 IDPS algorithm on Jun 22, 2015, 19:43 UTC, and they are also used in the new Enterprise VIIRS aerosol algorithm. "Move on to next internal test" means no actions are taken for the current internal test if any of the criteria are not met, and the algorithm continues to**

**the next internal test.**



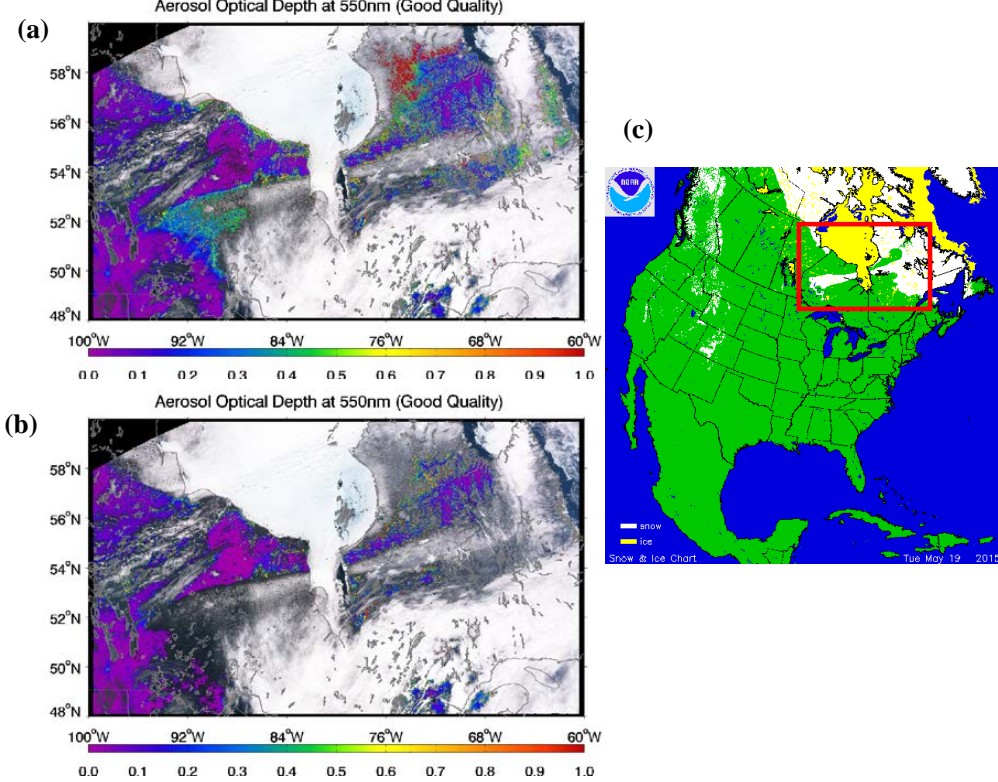

**Figure 2. VIIRS 'Good' quality AOT retrievals on May 19, 2015: (a) with the old VRA-based internal snow test; (b) with the new snow and snowmelt screening scheme; and (c) the snow cover map produced by the NOAA National Centers for Environmental Information (NCEI) (https://www.ncdc.noaa.gov/snow-and-ice/snow-cover/us/20150519). The red square in (c) shows the areas covered in (a) and (b). Note the snow cover to the east and south of the Hudson Bay in (c) and the associated anomalously high AOT values over the same areas in (a) but not in (b).**






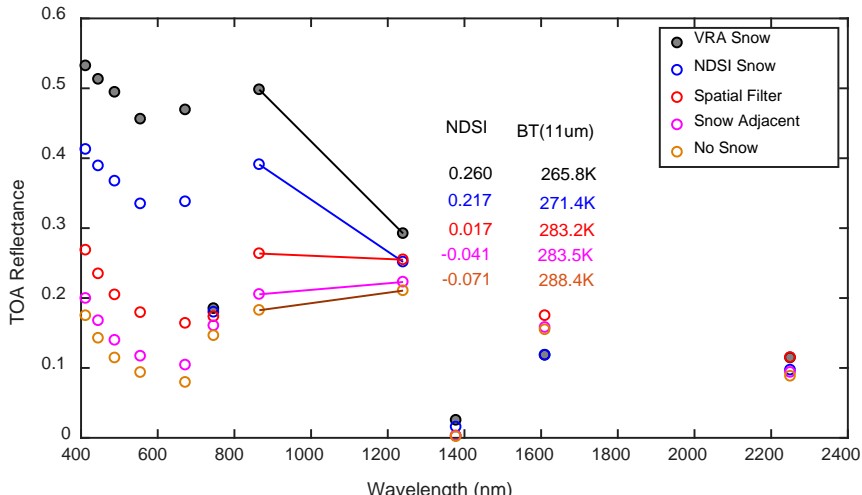

**Figure 3. Spectral curve of the TOA reflectance at 11 VIIRS wavelengths in five populations of pixels selected based on internal tests.**



(a)

(b)

(c)

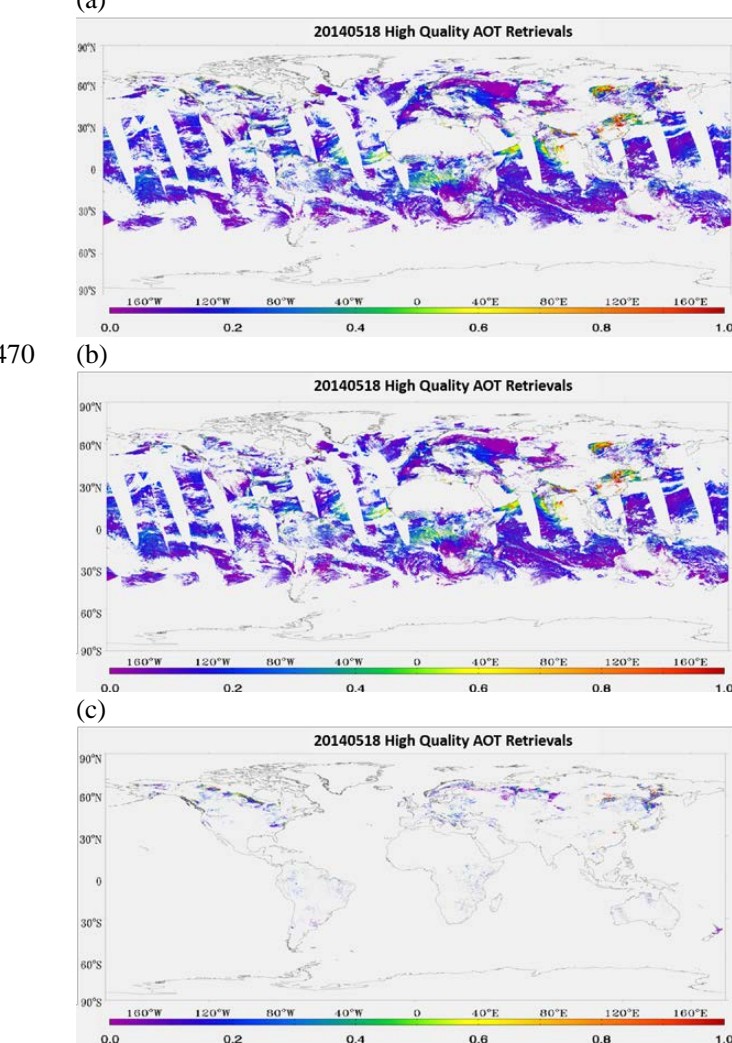

**Figure 4. The best quality VIIRS AOT retrievals on May 18, 2014: (a) with the old internal VRA-based snow test; (b) with the new NDSI-**
**based snow and snowmelt screening scheme; and (c) the difference between (b) and (a) that were either removed because of the new**
**snow test or were degraded by the new snow adjacency test and the spatial homogeneity test.**





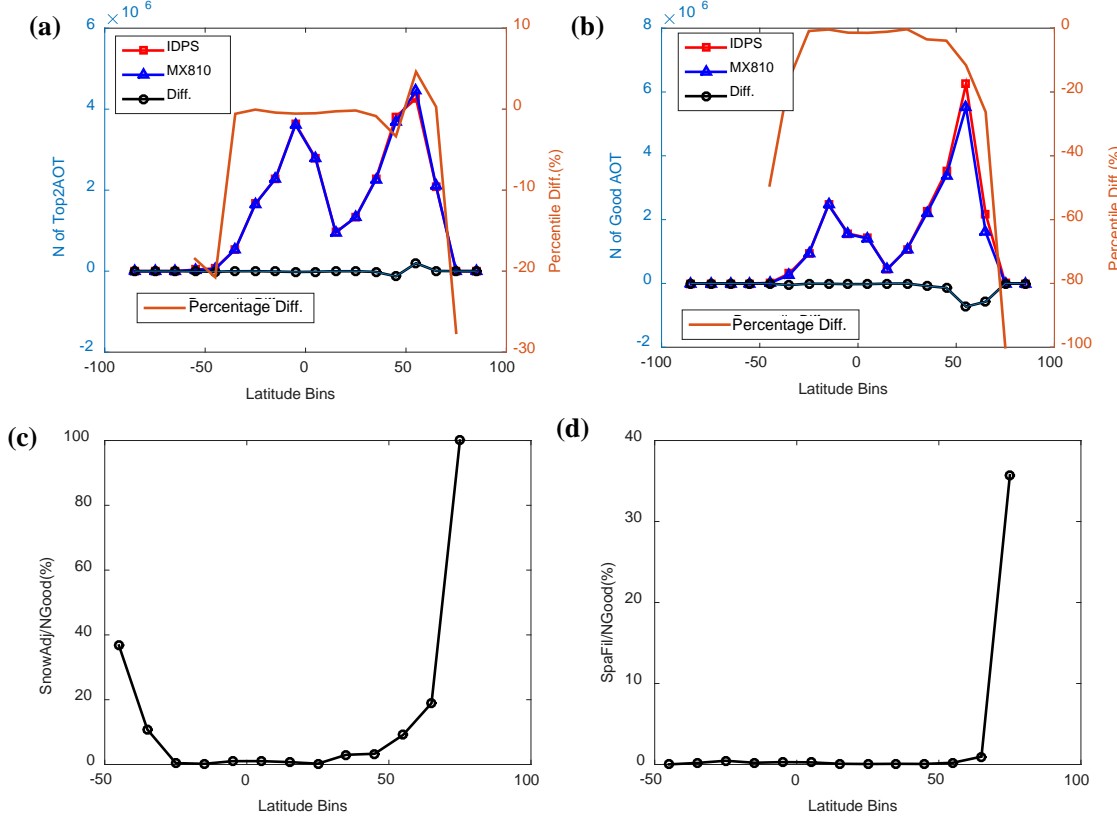


**Figure 5. Statistics in the sample size and percentage change in the AOT retrievals on May 18, 2014 as a function of 10° latitude bin: (a) Number of Good and Degraded quality AOT retrievals ("Top2AOT"), produced by the algorithm running the old VRA-based snow mask (red line with squares) and the new NDSI-based snow mask (blue line with triangles), and the difference between the two (black line with open circles). The brownish red line displays the difference as a percentage that is read along the right-hand axis. (b) Same as**

**(a) but for good quality AOT retrievals only; (c) the percentage of the pixels flagged as adjacent to an identified snow pixel over the total number of 'Good' quality AOT IP retrievals; and (d) the percentage of the pixels flagged by the homogeneity test (spatial filter) over the total number of 'Good' quality AOT retrievals. Top 2 quality in (a) includes both Good and Degraded quality AOT retrievals.**






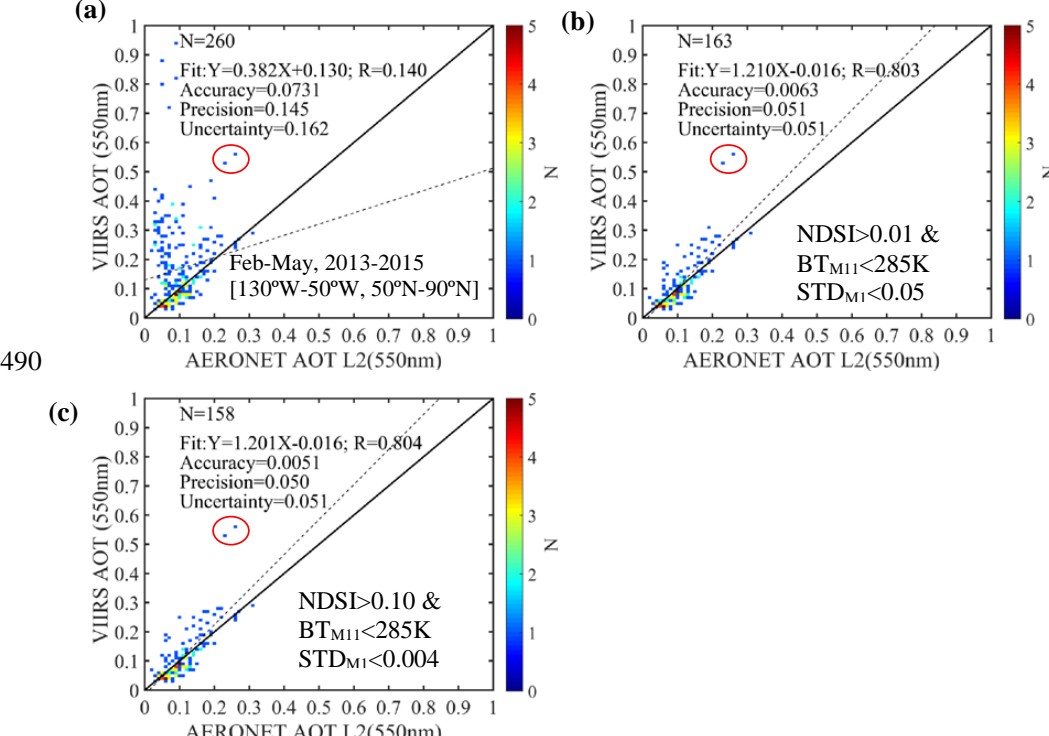

**Figure 6. Matchups of the VIIRS High Quality AOT retrievals with AERONET over [130ºW-50ºW, 50ºN-90ºN] during boreal spring Feb-May, 2013-2015: (a) with the old VRA-based internal snow test; (b) with the new snow and snowmelt screening scheme and the thresholds used in the Mx8.10 IDPS algorithm; and (c) with the new snow and snowmelt screening scheme but the thresholds used in the EPS algorithm. The two anomalous points with higher positive biases in the red circles were found to be more related to the AOT retrieval itself rather than snow under-screening.**





(a)

(b)


(c)

**Figure 7. S-NPP VIIRS seasonal mean of high quality AOT during the spring thaw season, March – May 2015: (a) with the old VRA-based snow test in the IDPS Aerosol Algorithm; (b) with the new snow and snowmelt screening scheme in the EPS Aerosol Algorithm; and (c) high quality AOT retrievals in the IDPS product but not in the EPS product, mainly attributable to the improved snow and snowmelt tests. The most outstanding snow and snowmelt under screening regions over northern hemisphere high latitude regions are highlighted in red squares. In the maps shown the original 750-m pixel-level AOT retrievals were mapped to 0.25 x 0.25 degree equal-angle latitude and longitude grids.**
