# Peer review of "Screening for Snow/Snowmelt in SNPP VIIRS Aerosol Optical Depth Algorithm"

_Atmospheric Measurement Techniques, 2018_

## Referee Comment (RC1) · Anonymous Referee #1 · 18 Jun 2018

This manuscript reports a development of an algorithm that effectively screens out snow/melting snow pixels in SNPP VIIRS aerosol optical depth product. The algorithm is described in details. Comparisons with AERONET measurements show that the algorithm works to effectively remove the snow and snow-melting pixels in the aerosol product. The paper is well written. Figures are sufficient and in general have good quality. Outcome of the study is of great importance to scientific studies that use the VIIRS aerosol product. I recommend the paper be published in the journal of AMT, before a few minor revisions are done.

1. While the "aerosol optical depth" (AOD) is used in title. the "aerosol optical thickness" (AOT) is used in abstract and main text. Although AOD and AOT have been used interchangeably in literature for quite a while, I would suggest a consistent use of

terminology throughout the paper. Furthermore, I prefer to use AOD.

2. Figure 4 has low quality, although other figures have good quality. Also, three panels appear to have the same title, which is a bit confusing. I would like to suggest that they re-plot the figure with high quality. For panel (c), a different color scale may be used to better represent the difference.

3. abstract, line 21: VRA appears too abruptly without any explanation. It may be suffice to say just the "default" snow-removing algorithm.

4. page 2, line 72: add "snow" immediately before "contamination".

―――――――――――――――――

---

## Referee Comment (RC2) · Anonymous Referee #3 · 20 Jun 2018

The article demonstrates that the VIIRS aerosol optical depth product contains snow/ice contamination issue over high latitude Northern hemisphere. New empirical snow and snowmelt masking was developed combining normalized differences snow index, brightness temperature threshold, snow adjacency test, and spatial homogeneity test. The impacts of the new masking were tested globally and validated against ground based AOD measurements. The topic is suitable for AMT and the contents are informative for the aerosol remote sensing community. The manuscript is well prepared. However, there are several issues that need to be addressed before this manuscript is suitable for publishing.

The manuscript indicated that there are two aerosol algorithm that applied on VIIRS sensor. It is not clear the reason to have two different coefficients of snow masks in

two algorithms. Does IDPS have similar problem of masking out thick haze using the C1=0.01? Can the snow mask of EPS be applied to IDPS?

Author discussed new snow mask for IDPS and EPS throughout the paper, however, in Fig. 2 the case study for EPS is missing. Without the case study the audience do know how under what boundary conditions the snow mask for EPS is different from the snow mask for IDPS. The author failed to explain how five populations of pixels were generated in Fig. 3. Figure 5 analyzes the data loss due to different masking procedures, which is very dependent on the topography, the snow distribution and such. Only use one day as an example is not statistically significant. Figure 5 concludes that there are additional 3.44% loss of data however, in Figure 6 there are 16% (43/260) data loss for data that are collocated with AERONET. The total data loss is 37% (97/260), which is much larger than the estimates from Fig. 5. Also in Fig. 5 there are different number of latitude bins after 50 degree north. It is not clear to me the physical meaning of snow adjacent percentage is 100%. It is more likely that at that latitude, there is no data available for this day. Also, the author does not mention the quality of data whether they are "Good" data or all quality data in Fig. 6. Although the discussion of Fig. 6 indicates only "Good" data are used in the analyses, but the author should clearly state it. The one last question is with the change of snow masking, what is the statistics of valid aerosol data that are misidentified as snow globally?

---

## Referee Comment (RC3) · Anonymous Referee #4 · 21 Jun 2018

This paper described the snow/snowmelt screening scheme for VIIRS AOD. The presentation is quite clear and the article is well-organized and concise. The method obviously works well as it has been implemented in the NOAA operation. It will be a useful documentation for the VIIRS AOD users.

Specific points:

Title: AOT is used throughout the manuscript. Why is aerosol optical DEPTH used in the title?

Line 49: The word "artificial" is redundant here.

Lines 76-78: It is stated that the snow screening tests "are designed to prevent the aerosol algorithm from making retrievals in inappropriate snow cover conditions" al-
though true Snow/Ice products are also available (Key et al., 2013). Can the authors comment on why the VIIRS Snow/Ice products are not used in the AOD algorithm?

Line 99-103: They are identical to those listed in Table 1. It is better not to repeat the same words. Same for lines 149-153, 176-178.

Line 153: has been set 0.01 in Mx8.10 and in newer versions -> has been set to 0.01 in Mx8.10 and newer versions

Line 170: Are the 7x7 area centered around the snow pixel?

Line 188: Something does not correctly show after the parenthesis.

Line 206: Please elaborate what are the criteria for the careful selection?

Lines 219-281: These two paragraphs are too long. Try breaking them into short ones.

Line 432: The sorting is not right here. This reference should be moved to Line 401.

Figure 1: Difficult to read. Please improve the quality of the image.

Figure 3: The three colors for the last three populations are too difficult to differentiate. Please change them to other distinct colors.

Figure 4: Difficult to read. Please improve the quality of the image.

Figure 5: Some "N of Good AOT" numbers in (b) are greater than "N of Top2AOT" in some latitude bins (e.g. lat=60). Something seems wrong here.

Figure 5: Showing "-100" and "100" for Latitude Bins should be avoided.

---

## Referee Comment (RC4) · Anonymous Referee #2 · 4 Jul 2018

This paper presents a modified NDSI-based snow detection scheme, which has been applied to the operational NOAA VIIRS IDPS and EPS aerosol algorithms. The proposed scheme effectively mitigated the snow contamination in AOD product by more accurately filter out pixels containing melting snow over high latitude regions. This is achieved by combining NDSI with various tests, such as brightness temperature, spatial variability, and spatial adjacency tests. Since snow contamination in the retrieval pixel even small amount could potentially lead to a significant high bias in AOD, it is important to implement more rigorous snow detection schemes in the aerosol algorithms and examine the impacts. The manuscript is well written and easy to follow. I believe that addressing the following comments would improve the quality of the paper further.

General comments 1. The proposed snow detection scheme consists of several steps.

[Figure]

I would recommend to extend Figure 2 and include AOD plots at every step, so that the readers can easily understand the impact of each step. The plot should probably be zoomed in more to better show the details. Plots of the test variables, i.e., NDSI, BT, and spatial variability, would help as well.

2. One can assume from Figure 4 that the proposed scheme results in some false alarm (snow detection in low latitudes, and low AOD in some snow-contaminated pixels). I would recommend to discuss this together with potential future work to further refine the scheme, as I think retaining good pixels is as important as removing bad pixels.

3. In Figure 6, I wonder if the three data points at AERONET AOD of ~0.05 and VIIRS AOD of ~0.2 are retrieval-related or snow-related.

Specific comments I don't find further specific comments other than the other reviewers'.

---

## Author Comment (AC1) · 19 Aug 2018

This manuscript reports a development of an algorithm that effectively screens out snow/melting snow pixels in SNPP VIIRS aerosol optical depth product. The algorithm is described in details. Comparisons with AERONET measurements show that the algorithm works to effectively remove the snow and snow-melting pixels in the aerosol product. The paper is well written. Figures are sufficient and in general have good quality. Outcome of the study is of great importance to scientific studies that use the VIIRS aerosol product. I recommend the paper be published in the journal of AMT, before a few minor revisions are done.

Thank you very much for your very valuable and constructive comments.

1. While the "aerosol optical depth" (AOD) is used in title the "aerosol optical thickness" (AOT) is used in abstract and main text. Although AOT and AOD have been used interchangeably in literature for quite a while, I would suggest a consistent use of terminology throughout the paper. Furthermore, I prefer to use AOD.

As suggested, the paper is revised for consistent use of terminology throughout. AOD is used to replace AOT in the paper.

2. Figure 4 has low quality, although other figures have good quality. Also, three panels appear to have the same title, which is a bit confusing. I would like to suggest that they re-plot the figure with high quality. For panel (c), a different color scale may be used to better represent the difference.
Figure 4 is now revised as suggested. High quality images are used with proper titles consistent with the captions.

3. abstract, line 21: VRA appears too abruptly without any explanation. It may be suffice to say just the "default" snow-removing algorithm.
We appreciate this comment. After some deliberation, considering the VRA method is introduced in details in the main text and the VRA-based snow test are referred in many places in the paper to compare with the NDSI-based tests, we think it is better to keep the 'VRA-based snow test' in the abstract rather than replace it with 'default'. This will avoid potential confusion to readers as well.

4. page 2, line 72: add "snow" immediately before "contamination"
Revised as suggested.

---

## Author Comment (AC2) · 19 Aug 2018

This paper presents a modified NDSI-based snow detection scheme, which has been applied to the operational NOAA VIIRS IDPS and EPS aerosol algorithms. The proposed scheme effectively mitigated the snow contamination in AOD product by more accurately filter out pixels containing melting snow over high latitude regions. This is achieved by combining NDSI with various tests, such as brightness temperature, spatial variability, and spatial adjacency tests. Since snow contamination in the retrieval pixel even small amount could potentially lead to a significant high bias in AOD, it is important to implement more rigorous snow detection schemes in the aerosol algorithms and examine the impacts. The manuscript is well written and easy to follow. I believe that addressing the following comments would improve the quality of the paper further.

Thank you very much for your very valuable comments that are extremely helpful to improving the quality of the paper.

General comments:
1. The proposed snow detection scheme consists of several steps. I would recommend to extend Figure 2 and include AOD plots at every step, so that the readers can easily understand the impact of each step. The plot should probably be zoomed in more to better show the details. Plots of the test variables, i.e., NDSI, BT, and spatial variability, would help as well.

Thanks for this valuable comment. The new snow and snowmelt scheme work as a whole for snow and snowmelt contamination removal, so we would think that a comparison between Figure 2(a) and 2(b) are sufficient to show the significant improvements in the new scheme from the old VRA based scheme, and at the same time, it avoids potential confusion to readers by adding too many subplots.
Meanwhile, we agree readers may want to know how the three components (NDSI test, snow adjacency test, and spatial filter) play their roles in the combined effect in the new scheme. Such information is much better shown in Figure 5 with more quantitative evaluation and discussions within Section 4.

2. One can assume from Figure 4 that the proposed scheme results in some false alarm (snow detection in low latitudes, and low AOD in some snow-contaminated pixels). I would

recommend to discuss this together with potential future work to further refine the scheme, as I think retaining good pixels is as important as removing bad pixels.

This is a very good point. In the algorithm development, in addition to avoid snow and cloud contamination, we strive to avoid over-screening as well. In Section 3, we particularly discussed the complement impact of the spatial filter that it also effectively screens low level 'popcorn' cumulus clouds at low latitude regions. Since both cloud and snow conditions are unfavorable conditions for meaningful satellite aerosol retrievals, the homogeneity test provides additional quality assurances to the VIIRS aerosol retrievals in terms of both snow and cloud screenings. However we agree that residual false alarm may still remain even after the snow/snowmelt screening scheme is updated. It is a daunting challenge to verify whether the low AOD in some snow-contaminated pixels are real AOD signals or contaminated by snowmelt conditions that are not necessarily causing high AOD retrievals like snow conditions. We are adding more discussion on Page 9 in Section 5 that the algorithm should be further improved in future work: "In future work, in order to reach more quantitative statistics for a better understanding of the relative contributions from each test, more testing dates at different seasons are needed. The additional testing will not only help find seasonal variability of the tests, but also help identify any residual snow and snowmelt contaminations or any over-screened AOD retrievals, both of which are valuable for further algorithm improvement. "

3. In Figure 6, I wonder if the three data points at AERONET AOD of ~0.05 and VIIRS AOD of~0.2 are retrieval-related or snow-related.

On Line 277-282, we have discussed the two points in the red circles are retrieval-related, which was verified by our additional testing runs with the new EPS algorithm.
For the additional three points at AERONET AOD of ~0.05 and VIIRS AOD of ~0.2, we agree with the reviewer that it seems the new snow/snowmelt scheme did not screen the matchup out as snow/snowmelt contamination. Given our confidence on the performance of the new scheme, we believe the remaining bias (~0.15) are not snow or snowmelt related, and the aerosol algorithm should be improved to further reduce the retrieval bias and data uncertainty.

Specific comments I don't find further specific comments other than the other reviewers'.

Thanks.

---

## Author Comment (AC3) · 19 Aug 2018

The article demonstrates that the VIIRS aerosol optical depth product contains snow/ice contamination issue over high latitude Northern hemisphere. New empirical snow and snowmelt masking was developed combining normalized differences snow index, brightness temperature threshold, snow adjacency test, and spatial homogeneity test. The impacts of the new masking were tested globally and validated against ground based AOD measurements. The topic is suitable for AMT and the contents are informative for the aerosol remote sensing community. The manuscript is well prepared. However, there are several issues that need to be addressed before this manuscript is suitable for publishing.

Thank you very much for your very detailed and thoughtful comments. Your suggestions are very valuable for us to further improve the quality of the paper. Please see below for our responses highlighted in blue. Thanks.

The manuscript indicated that there are two aerosol algorithm that applied on VIIRS sensor. It is not clear the reason to have two different coefficients of snow masks in two algorithms. Does IDPS have similar problem of masking out thick haze using the C1=0.01? Can the snow mask of EPS be applied to IDPS?

On Page 2 Line 63 – 71, we introduced the evolving IDPS aerosol algorithms and the new EPS aerosol algorithm that will replace the IDPS aerosol algorithm. The new EPS algorithm is different from the IDPS algorithm in many ways such as new AOD retrieval techniques and new screening schemes etc. If C1=0.01 is used, both IDPS and EPS aerosol products will have the same 'thick haze over-screening' issue. Since the IDPS aerosol product will be replaced by the EPS aerosol product, we only adjusted the C1 value for the EPS algorithm for testing purposes of the new snow/snowmelt scheme.

In our previous response to reviewers' comments, we also had addressed this same concern before the AMTD publication:
"Because of the newly discovered over-screened thick haze issue that is attributable to the snow/snowmelt over-screening, the new snow mask was further refined by tuning threshold values, and it has been implemented in the NOAA Enterprise Processing System (EPS) VIIRS Aerosol Algorithm. Although both algorithms are currently running

operationally, one at IDPS and the other at NDE, the EPS aerosol algorithm will eventually replace the IDPS algorithm, therefore we are not seeking to further improve the snow mask in the IDPS aerosol algorithm any more. Instead, the S-NPP VIIRS aerosol products will be reprocessed by the new EPS algorithm."

Author discussed new snow mask for IDPS and EPS throughout the paper, however, in Fig. 2 the case study for EPS is missing. Without the case study the audience do know how under what boundary conditions the snow mask for EPS is different from the snow mask for IDPS.

In our previous responses to the reviewers' comment before the AMTD, we had addressed this same concern. For the aerosol retrievals in Figure 2, the EPS retrieval is very similar to the IDPS retrieval after the snow screening is updated.

The author failed to explain how five populations of pixels were generated in Fig. 3.

The explanation of five populations of pixels in Figure 3 are provided on Page 7 Line 227-230, followed by more discussions on the Figure from Lines 231-246.

Figure 5 analyzes the data loss due to different masking procedures, which is very dependent on the topography, the snow distribution and such. Only use one day as an example is not statistically significant.

Snow screening issue is more significant over boreal spring season and we choose spring dates to highlight the issue. Figure 5 demonstrates an example that is typical for spring days but not for global annual average conditions.

Figure 5 concludes that there are additional 3.44% loss of data however, in Figure 6 there are 16% (43/260) data loss for data that are collocated with AERONET. The total data loss is 37% (97/260), which is much larger than the estimates from Fig. 5.

There are fundamental difference between Figure 5 and Figure 6 statistics. Figure 5 is global evaluation but Figure 6 only counts VIIRS vs. AERONET match ups. For Figure 6, we only selected Northern North America as our region of interest, and selected boreal spring time from March to May as highlight seasons. Therefore the data loss is much larger than global evaluation in Figure 5.

Also in Fig. 5 there are different number of latitude bins after 50 degree north. It is not clear to me the physical meaning of snow adjacent percentage is 100%. It is more likely that at that latitude, there is no data available for this day.

We use 10 degree latitude bins for all figures in Figure 5. Because aerosol retrievals are only available over snow free regions, aerosol retrievals over high latitudes are very limited. Taking Figure 5c for example, there were no aerosol retrievals when latitude are higher than 75 degrees. For the 60-70 degree bin, the 100% indicates that those old retrievals that were previously contaminated because of old snow screening are now removed after the snow screening methods are updated.

Also, the author does not mention the quality of data whether they are "Good" data or all quality data in Fig. 6. Although the discussion of Fig. 6 indicates only "Good" data are used in the analyses, but the author should clearly state it.

We added 'Good Quality' in Page 7 Line 263 and in the Figure 6 caption as well. We only use good quality VIIRS retrievals for validation purposes. Thanks a lot for pointing this out.

The one last question is with the change of snow masking, what is the statistics of valid aerosol data that are misidentified as snow globally?

On Page 9 Line 313 to 316, for a global testing on May 18, 2014, a typical day in spring thaw season when snow and snowmelt prevail, the new snow test screened out an additional 3.44% 'Good' quality VIIRS AOD retrievals, which were otherwise contaminated by snow and snowmelt. This means if the snow screens are not updated, we likely have 3.44% valid aerosol data that are misidentified as snow globally for a typical data in spring thaw season. This number is lower for other seasons when snow and snowmelt are not a significant issue for VIIRS aerosol retrievals.

---

## Author Comment (AC4) · 19 Aug 2018

This paper described the snow/snowmelt screening scheme for VIIRS AOD. The presentation is quite clear and the article is well-organized and concise. The method obviously works well as it has been implemented in the NOAA operation. It will be a useful documentation for the VIIRS AOD users.

Thank you very much for your very valuable comments and kind encouragement. Please see below for our responses highlighted in blue. Thanks.

Specific points:

Title: AOT is used throughout the manuscript. Why is aerosol optical DEPTH used in the title?

We have revised the paper to use AOD consistently throughout the paper.

Line 49: The word "artificial" is redundant here.

Removed as suggested.

Lines 76-78: It is stated that the snow screening tests "are designed to prevent the aerosol algorithm from making retrievals in inappropriate snow cover conditions" although true Snow/Ice products are also available (Key et al., 2013). Can the authors comment on why the VIIRS Snow/Ice products are not used in the AOD algorithm?

There could be two significant reasons. The first is the operational consideration. The operational VIIRS algorithms run in a chain. In this chain the operational snow/ice algorithm is downstream of the operational aerosol algorithm and thus the snow/ice retrievals are not yet available when the aerosol algorithm runs. The second reason is some consideration related to how the snow/ice product could be used if it was upstream before aerosol retrievals. The requirements of the snow/snowmelt contamination screening in the aerosol algorithm may be more conservative than the general snow detection in the snow/ice product. For example, we would prefer the aerosol algorithm does not retrieve

AOD over pixels with sub-pixel snow while the snow/ice product may not have the exact information of sub-pixel snow existence.

Line 99-103: They are identical to those listed in Table 1. It is better not to repeat the same words. Same for lines 149-153, 176-178.
These lines are only part of the information in Table 1. Table 1 also collects information such as AOD quality criteria and additional notes. The summary of this information in Table 1 provides organized information and better reference for readers.

Line 153: has been set 0.01 in Mx8.10 and in newer versions -> has been set to 0.01 in Mx8.10 and newer versions

Revised as suggested.

Line 170: Are the 7x7 area centered around the snow pixel?

Yes. As described in Line 172, the new snow adjacency test loops through the adjacent 7×7 pixels surrounding the central snow pixel.

Line 188: Something does not correctly show after the parenthesis.
It is $\rho_{412}$. It is corrected now.

Line 206: Please elaborate what are the criteria for the careful selection?
Since the test is threshold based, the unavoidable fact is, we have to achieve a balance between screening for sub-pixel snow and allowing retrievals of heavy smog. We tested different threshold values of C2, and determined C2=0.004 is optimal for minimizing sub-pixel snow over-screening to allow reasonable heavy smog retrievals at the same time.

Lines 219-281: These two paragraphs are too long. Try breaking them into short ones.
Revised as suggested. The two paragraphs are now broken into several short paragraphs.

Line 432: The sorting is not right here. This reference should be moved to Line 401.
Corrected.

Figure 1: Difficult to read. Please improve the quality of the image.
The images were replaced with high resolution ones as suggested to improve their quality.

Figure 3: The three colors for the last three populations are too difficult to differentiate. Please change them to other distinct colors.
Colors for the last three data populations are changed as suggested.

Figure 4: Difficult to read. Please improve the quality of the image.
The quality of the image is improved as suggested.

Figure 5: Some "N of Good AOD" numbers in (b) are greater than "N of Top2AOD" in some latitude bins (e.g. lat=60). Something seems wrong here.
Thank you very much for pointing out this error to us. Figure 5(a) should be 'Degraded' Quality AOD instead of 'Top 2' AOD. Top 2 AOD retrievals includes both Good and Degraded quality AOD retrievals. We have revised the Y axis label in (a) from 'Top 2' to 'Degraded' and corrected the Figure caption accordingly. The corresponding discussions are also updated.

Figure 5: Showing "-100" and "100" for Latitude Bins should be avoided.

"-100" and "100" are removed and the x-axis labels are revised as suggested.

---

## Author Response (AR1)

This manuscript reports a development of an algorithm that effectively screens out snow/melting snow pixels in SNPP VIIRS aerosol optical depth product. The algorithm is described in details. Comparisons with AERONET measurements show that the algorithm works to effectively remove the snow and snow-melting pixels in the aerosol product. The paper is well written. Figures are sufficient and in general have good quality. Outcome of the study is of great importance to scientific studies that use the VIIRS aerosol product. I recommend the paper be published in the journal of AMT, before a few minor revisions are done.

Thank you very much for your very valuable and constructive comments.

1. While the "aerosol optical depth" (AOD) is used in title the "aerosol optical thickness" (AOT) is used in abstract and main text. Although AOT and AOD have been used interchangeably in literature for quite a while, I would suggest a consistent use of terminology throughout the paper. Furthermore, I prefer to use AOD.

As suggested, the paper is revised for consistent use of terminology throughout. AOD is used to replace AOT in the paper.

2. Figure 4 has low quality, although other figures have good quality. Also, three panels appear to have the same title, which is a bit confusing. I would like to suggest that they re-plot the figure with high quality. For panel (c), a different color scale may be used to better represent the difference.
Figure 4 is now revised as suggested. High quality images are used with proper titles consistent with the captions.

3. abstract, line 21: VRA appears too abruptly without any explanation. It may be suffice to say just the "default" snow-removing algorithm.
We appreciate this comment. After some deliberation, considering the VRA method is introduced in details in the main text and the VRA-based snow test are referred in many places in the paper to compare with the NDSI-based tests, we think it is better to keep the 'VRA-based snow test' in the abstract rather than replace it with 'default'. This will avoid potential confusion to readers as well.

4. page 2, line 72: add "snow" immediately before "contamination"
Revised as suggested.

This paper presents a modified NDSI-based snow detection scheme, which has been applied to the operational NOAA VIIRS IDPS and EPS aerosol algorithms. The proposed scheme effectively mitigated the snow contamination in AOD product by more accurately filter out pixels containing melting snow over high latitude regions. This is achieved by combining NDSI with various tests, such as brightness temperature, spatial variability, and spatial adjacency tests. Since snow contamination in the retrieval pixel even small amount could potentially lead to a significant high bias in AOD, it is important to implement more rigorous snow detection schemes in the aerosol algorithms and examine the impacts. The manuscript is well written and easy to follow. I believe that addressing the following comments would improve the quality of the paper further.

Thank you very much for your very valuable comments that are extremely helpful to improving the quality of the paper.

General comments:
1. The proposed snow detection scheme consists of several steps. I would recommend to extend Figure 2 and include AOD plots at every step, so that the readers can easily understand the impact of each step. The plot should probably be zoomed in more to better show the details. Plots of the test variables, i.e., NDSI, BT, and spatial variability, would help as well.

Thanks for this valuable comment. The new snow and snowmelt scheme work as a whole for snow and snowmelt contamination removal, so we would think that a comparison between Figure 2(a) and 2(b) are sufficient to show the significant improvements in the new scheme from the old VRA based scheme, and at the same time, it avoids potential confusion to readers by adding too many subplots.
Meanwhile, we agree readers may want to know how the three components (NDSI test, snow adjacency test, and spatial filter) play their roles in the combined effect in the new scheme. Such information is much better shown in Figure 5 with more quantitative evaluation and discussions within Section 4.

2. One can assume from Figure 4 that the proposed scheme results in some false alarm (snow detection in low latitudes, and low AOD in some snow-contaminated pixels). I would

recommend to discuss this together with potential future work to further refine the scheme, as I think retaining good pixels is as important as removing bad pixels.

This is a very good point. In the algorithm development, in addition to avoid snow and cloud contamination, we strive to avoid over-screening as well. In Section 3, we particularly discussed the complement impact of the spatial filter that it also effectively screens low level 'popcorn' cumulus clouds at low latitude regions. Since both cloud and snow conditions are unfavorable conditions for meaningful satellite aerosol retrievals, the homogeneity test provides additional quality assurances to the VIIRS aerosol retrievals in terms of both snow and cloud screenings. However we agree that residual false alarm may still remain even after the snow/snowmelt screening scheme is updated. It is a daunting challenge to verify whether the low AOD in some snow-contaminated pixels are real AOD signals or contaminated by snowmelt conditions that are not necessarily causing high AOD retrievals like snow conditions. We are adding more discussion on Page 9 in Section 5 that the algorithm should be further improved in future work: "In future work, in order to reach more quantitative statistics for a better understanding of the relative contributions from each test, more testing dates at different seasons are needed. The additional testing will not only help find seasonal variability of the tests, but also help identify any residual snow and snowmelt contaminations or any over-screened AOD retrievals, both of which are valuable for further algorithm improvement.   "

3. In Figure 6, I wonder if the three data points at AERONET AOD of ~0.05 and VIIRS AOD of~0.2 are retrieval-related or snow-related.

On Line 277-282, we have discussed the two points in the red circles are retrieval-related, which was verified by our additional testing runs with the new EPS algorithm.
For the additional three points at AERONET AOD of ~0.05 and VIIRS AOD of ~0.2, we agree with the reviewer that it seems the new snow/snowmelt scheme did not screen the matchup out as snow/snowmelt contamination. Given our confidence on the performance of the new scheme, we believe the remaining bias (~0.15) are not snow or snowmelt related, and the aerosol algorithm should be improved to further reduce the retrieval bias and data uncertainty.

Specific comments I don't find further specific comments other than the other reviewers'.

Thanks.

The article demonstrates that the VIIRS aerosol optical depth product contains snow/ice contamination issue over high latitude Northern hemisphere. New empirical snow and snowmelt masking was developed combining normalized differences snow index, brightness temperature threshold, snow adjacency test, and spatial homogeneity test. The impacts of the new masking were tested globally and validated against ground based AOD measurements. The topic is suitable for AMT and the contents are informative for the aerosol remote sensing community. The manuscript is well prepared. However, there are several issues that need to be addressed before this manuscript is suitable for publishing.

Thank you very much for your very detailed and thoughtful comments. Your suggestions are very valuable for us to further improve the quality of the paper. Please see below for our responses highlighted in blue. Thanks.

The manuscript indicated that there are two aerosol algorithm that applied on VIIRS sensor. It is not clear the reason to have two different coefficients of snow masks in two algorithms. Does IDPS have similar problem of masking out thick haze using the C1=0.01? Can the snow mask of EPS be applied to IDPS?

On Page 2 Line 63 – 71, we introduced the evolving IDPS aerosol algorithms and the new EPS aerosol algorithm that will replace the IDPS aerosol algorithm. The new EPS algorithm is different from the IDPS algorithm in many ways such as new AOD retrieval techniques and new screening schemes etc. If C1=0.01 is used, both IDPS and EPS aerosol products will have the same 'thick haze over-screening' issue. Since the IDPS aerosol product will be replaced by the EPS aerosol product, we only adjusted the C1 value for the EPS algorithm for testing purposes of the new snow/snowmelt scheme.

In our previous response to reviewers' comments, we also had addressed this same concern before the AMTD publication:
"Because of the newly discovered over-screened thick haze issue that is attributable to the snow/snowmelt over-screening, the new snow mask was further refined by tuning threshold values, and it has been implemented in the NOAA Enterprise Processing System (EPS) VIIRS Aerosol Algorithm. Although both algorithms are currently running

operationally, one at IDPS and the other at NDE, the EPS aerosol algorithm will eventually replace the IDPS algorithm, therefore we are not seeking to further improve the snow mask in the IDPS aerosol algorithm any more. Instead, the S-NPP VIIRS aerosol products will be reprocessed by the new EPS algorithm."

Author discussed new snow mask for IDPS and EPS throughout the paper, however, in Fig. 2 the case study for EPS is missing. Without the case study the audience do know how under what boundary conditions the snow mask for EPS is different from the snow mask for IDPS.

In our previous responses to the reviewers' comment before the AMTD, we had addressed this same concern. For the aerosol retrievals in Figure 2, the EPS retrieval is very similar to the IDPS retrieval after the snow screening is updated.

The author failed to explain how five populations of pixels were generated in Fig. 3.

The explanation of five populations of pixels in Figure 3 are provided on Page 7 Line 227-230, followed by more discussions on the Figure from Lines 231-246.

Figure 5 analyzes the data loss due to different masking procedures, which is very dependent on the topography, the snow distribution and such. Only use one day as an example is not statistically significant.

Snow screening issue is more significant over boreal spring season and we choose spring dates to highlight the issue. Figure 5 demonstrates an example that is typical for spring days but not for global annual average conditions.

Figure 5 concludes that there are additional 3.44% loss of data however, in Figure 6 there are 16% (43/260) data loss for data that are collocated with AERONET. The total data loss is 37% (97/260), which is much larger than the estimates from Fig. 5.

There are fundamental difference between Figure 5 and Figure 6 statistics. Figure 5 is global evaluation but Figure 6 only counts VIIRS vs. AERONET match ups. For Figure 6, we only selected Northern North America as our region of interest, and selected boreal spring time from March to May as highlight seasons. Therefore the data loss is much larger than global evaluation in Figure 5.

Also in Fig. 5 there are different number of latitude bins after 50 degree north. It is not clear to me the physical meaning of snow adjacent percentage is 100%. It is more likely that at that latitude, there is no data available for this day.

We use 10 degree latitude bins for all figures in Figure 5. Because aerosol retrievals are only available over snow free regions, aerosol retrievals over high latitudes are very limited. Taking Figure 5c for example, there were no aerosol retrievals when latitude are higher than 75 degrees. For the 60-70 degree bin, the 100% indicates that those old retrievals that were previously contaminated because of old snow screening are now removed after the snow screening methods are updated.

Also, the author does not mention the quality of data whether they are "Good" data or all quality data in Fig. 6. Although the discussion of Fig. 6 indicates only "Good" data are used in the analyses, but the author should clearly state it.

We added 'Good Quality' in Page 7 Line 263 and in the Figure 6 caption as well. We only use good quality VIIRS retrievals for validation purposes. Thanks a lot for pointing this out.

The one last question is with the change of snow masking, what is the statistics of valid aerosol data that are misidentified as snow globally?

On Page 9 Line 313 to 316, for a global testing on May 18, 2014, a typical day in spring thaw season when snow and snowmelt prevail, the new snow test screened out an additional 3.44% 'Good' quality VIIRS AOD retrievals, which were otherwise contaminated by snow and snowmelt. This means if the snow screens are not updated, we likely have 3.44% valid aerosol data that are misidentified as snow globally for a typical data in spring thaw season. This number is lower for other seasons when snow and snowmelt are not a significant issue for VIIRS aerosol retrievals.

This paper described the snow/snowmelt screening scheme for VIIRS AOD. The presentation is quite clear and the article is well-organized and concise. The method obviously works well as it has been implemented in the NOAA operation. It will be a useful documentation for the VIIRS AOD users.

Thank you very much for your very valuable comments and kind encouragement. Please see below for our responses highlighted in blue. Thanks.

Specific points:

Title: AOT is used throughout the manuscript. Why is aerosol optical DEPTH used in the title?

We have revised the paper to use AOD consistently throughout the paper.

Line 49: The word "artificial" is redundant here.

Removed as suggested.

Lines 76-78: It is stated that the snow screening tests "are designed to prevent the aerosol algorithm from making retrievals in inappropriate snow cover conditions" although true Snow/Ice products are also available (Key et al., 2013). Can the authors comment on why the VIIRS Snow/Ice products are not used in the AOD algorithm?

There could be two significant reasons. The first is the operational consideration. The operational VIIRS algorithms run in a chain. In this chain the operational snow/ice algorithm is downstream of the operational aerosol algorithm and thus the snow/ice retrievals are not yet available when the aerosol algorithm runs. The second reason is some consideration related to how the snow/ice product could be used if it was upstream before aerosol retrievals. The requirements of the snow/snowmelt contamination screening in the aerosol algorithm may be more conservative than the general snow detection in the snow/ice product. For example, we would prefer the aerosol algorithm does not retrieve

AOD over pixels with sub-pixel snow while the snow/ice product may not have the exact information of sub-pixel snow existence.

Line 99-103: They are identical to those listed in Table 1. It is better not to repeat the same words. Same for lines 149-153, 176-178.
These lines are only part of the information in Table 1. Table 1 also collects information such as AOD quality criteria and additional notes. The summary of this information in Table 1 provides organized information and better reference for readers.

Line 153: has been set 0.01 in Mx8.10 and in newer versions -> has been set to 0.01 in Mx8.10 and newer versions

Revised as suggested.

Line 170: Are the 7x7 area centered around the snow pixel?

Yes. As described in Line 172, the new snow adjacency test loops through the adjacent 7×7 pixels surrounding the central snow pixel.

Line 188: Something does not correctly show after the parenthesis.
It is $\rho_{412}$. It is corrected now.

Line 206: Please elaborate what are the criteria for the careful selection?
Since the test is threshold based, the unavoidable fact is, we have to achieve a balance between screening for sub-pixel snow and allowing retrievals of heavy smog. We tested different threshold values of C2, and determined C2=0.004 is optimal for minimizing sub-pixel snow over-screening to allow reasonable heavy smog retrievals at the same time.

Lines 219-281: These two paragraphs are too long. Try breaking them into short ones.
Revised as suggested. The two paragraphs are now broken into several short paragraphs.

Line 432: The sorting is not right here. This reference should be moved to Line 401.
Corrected.

Figure 1: Difficult to read. Please improve the quality of the image.
The images were replaced with high resolution ones as suggested to improve their quality.

Figure 3: The three colors for the last three populations are too difficult to differentiate. Please change them to other distinct colors.
Colors for the last three data populations are changed as suggested.

Figure 4: Difficult to read. Please improve the quality of the image.
The quality of the image is improved as suggested.

Figure 5: Some "N of Good AOD" numbers in (b) are greater than "N of Top2AOD" in some latitude bins (e.g. lat=60). Something seems wrong here.
Thank you very much for pointing out this error to us. Figure 5(a) should be 'Degraded' Quality AOD instead of 'Top 2' AOD. Top 2 AOD retrievals includes both Good and Degraded quality AOD retrievals. We have revised the Y axis label in (a) from 'Top 2' to 'Degraded' and corrected the Figure caption accordingly. The corresponding discussions are also updated.

Figure 5: Showing "-100" and "100" for Latitude Bins should be avoided.

"-100" and "100" are removed and the x-axis labels are revised as suggested.

[revised manuscript text omitted]

**(a)**

[Figure]

20140518 VIIRS Best Quality AOD with VRA-based Snow Test

**(b)** 20140518 VIIRS Best Quality AOD with NDSI-based Snow and Snowmelt Tests

[Figure]

**(c)** 20140518 VIIRS Best Quality AOD in (a) Screened in (b)

[Figure]

[Figure]

480 (b)

[Figure]

(c)

[Figure]

485 **Figure 4. The best quality VIIRS AOD retrievals on May 18, 2014: (a) with the old internal VRA-based snow test; (b) with the new NDSI-based snow and snowmelt screening scheme; and (c) the difference between (b) and (a) that were either removed because of the new snow test or were degraded by the new snow adjacency test and the spatial homogeneity test.**

[Figure]

490

Figure 5. Statistics in the sample size and percentage change in the AOD retrievals on May 18, 2014 as a function of 10° latitude
bin: (a) Number of  Degraded' quality AOD retrievals , produced by the algorithm running the old VRA-
495  based snow mask (red line with squares) and the new NDSI-based snow mask (blue line with triangles), and the difference between the
two (black line with open circles). The brownish red line displays the difference as a percentage that is read along the right-hand axis.
(b) Same as (a) but for  Good' quality AOD retrievals only; (c) the percentage of the pixels flagged as adjacent to an identified
snow pixel over the total number of 'Good' quality AOD IP retrievals; and (d) the percentage of the pixels flagged by the
homogeneity test (spatial filter) over the total number of 'Good' quality AOD retrievals. ~~Top 2 quality in (a) includes both Good
and Degraded quality AOT retrievals.~~

500

[Figure]

**Figure 6.** Matchups of the VIIRS  Good Quality AOD retrievals with AERONET over [130ºW-50ºW, 50ºN-90ºN] during boreal spring Feb-May, 2013-2015: (a) with the old VRA-based internal snow test; (b) with the new snow and snowmelt screening scheme and the thresholds used in the Mx8.10 IDPS algorithm; and (c) with the new snow and snowmelt screening scheme but the thresholds used in the EPS algorithm. The two anomalous points with higher positive biases in the red circles were found to be more related to the AOD retrieval itself rather than snow under-screening.

505

**(a)**

[Figure]

510

**(b)**

[Figure]

**(c)**

[Figure]

515

**Figure 7. S-NPP VIIRS seasonal mean of  good quality AOD during the spring thaw season, March – May 2015: (a) with the old VRA-based snow test in the IDPS Aerosol Algorithm; (b) with the new snow and snowmelt screening scheme in the EPS Aerosol Algorithm; and (c)  good quality AOD retrievals in the IDPS product but not in the EPS product, mainly attributable to the improved snow and snowmelt tests. The most outstanding snow and snowmelt under screening regions over northern hemisphere high latitude regions are highlighted in red squares. In the maps shown the original 750-m pixel-level AOD retrievals were mapped to 0.25 x 0.25 degree equal-angle latitude and longitude grids.**

520